# A Ghost-Cell Immersed Boundary Method for Wave–Structure Interaction using a Two-Phase Flow Model

**Yuan-Shiang Tsai [1]** and **Der-Chang Lo [2],\***

[1] Master's Program in Offshore Wind Energy Engineering, College of Maritime, National Kaohsiung University of Science and Technology, Kaohsiung 80543, Taiwan; ystsai@nkust.edu.tw

[2] Department of Maritime Information and Technology, National Kaohsiung University of Science and Technology, Kaohsiung 80543, Taiwan

\* Correspondence: loderg@nkust.edu.tw; Tel.: +886-7-810-0888 (ext. 25320)

**Abstract:** The air-water two-phase flow model is developed to study the transformation of monochromatic waves passing over the submerged structure. The level set method is employed to describe the motion of the interface while the effect of the immersed object on the fluid is resolved using the ghost-cell immersed boundary method. The computational domain integrated with the air-water and fluid-solid phases allows the use of uniform Cartesian grids. The model simulates the wave generation, wave decomposition over a submerged trapezoidal breakwater, and the formation of the vortices as well as the drag and lift forces caused by the surface waves over three different configurations of the submerged structures. The numerical results show the capability of the present model to accurately track the deformation of the free surface. In addition, the variation of the drag and lift forces depend on the wavelength and wave induced vortices around the submerged object. Hence, the study observes that the triangular structure experiences the relatively small wave force.

**Keywords:** ghost-cell immersed boundary method; water wave transformation; wave-structure interactions; two-phase flows

## 1. Introduction

There are great demands to protect coast from the impact of ocean wave forces. This is mostly considered for locations with high population density such as the western Taiwan where many human and industrial activities are present and close to shores. In addition, climate change makes the sea level rise and produces extreme meteorological events such as the increase of typhoon intensity and hence formation of large waves. Coastal protection has become essential to prevent the loss of life and damage of economy [1–3]. A submerged breakwater is commonly constructed to diminish significant wave loadings, which is also in advantage of preserving natural features. Hence, understanding the wave evolution over submerged obstacles and the resultant flow fields to characterize the dynamic response in the wave-structure interaction is important to gain the knowledge for the design of breakwaters in terms of safety considerations.

For surface waves traveling over the submerged body, the physical process considers the incident waves propagate along with downstream distance, which are deformed by the shallow water and subsequently transmitted into the deeper water region. It has been known from direct field and laboratory observations that the interaction between the waves and submerged structure contributes to the generation and growth of higher harmonic waves [4–7]. In the experiment of Beji and Battjes and Luth et al. [6,7], the formation of higher harmonic waves was caused by the wave shoaling on the

top surface of the bar with shallow water and the transformation lasted with the waves moving away from the submerged structure and into the deeper water region.

With the improvement of numerical algorithms and rapid advances in computing technology, modeling of wave tanks to generate progressive waves has been widely used to study the transformation of the waves over submerged structures. Various numerical models have been developed, for example, the nonlinear shallow water wave equations, Boussinesq type equations, nonlinear inviscid Laplace's equation and Navier-Stokes equations [8–14]. It is likely that the Navier-Stokes equation with the momentum source wave-maker predicts the wave deformation well and velocity distribution close to the obstacle. Huang and Dong [13] numerically solved the 2D unsteady Navier-Stokes equations to simulate the wave deformation and vortex generation for propagative waves traveling over a submerged dike, for which the incident waves were generated using a piston-type wave maker. Based on the model of Huang and Dong [13], the numerical experiment of Huang and Dong [14] studied the solitary wave interacting with the submerged rectangle dike. Kawasaki [15] proposed the model in conjunction with volume of fluid (VOF) method to investigate the wave breaking over the submerged breakwater with various configurations.

The immersed boundary (IB) method is a powerful technique to effectively solve problems with the embedded boundary in the fluid. The approach directly introduces a body force to the momentum equations in advantage of neglecting the mapping procedures to simulate the force of the immersed body on the fluid flow. The IB method is originally developed by Peskin [16] to study cardiac mechanics associated with the blood flow. Important development was made by Goldstein et al. [17] and Saiki and Bringen [18] to employ an imposed force to the boundary condition at the immersed boundary. The approach has attracted a great deal of interest to simulate the fluid flows with embedded bodies in various fields, for example, the prediction of turbulent flow around the solid cylinder [19], development of ghost-cell immersed boundary method (GCIBM) to model the complex geometries over a wavy boundary and a three dimensional bump [20], combination of Lattice Boltzmann and IB method to study the transport and deformation of capsules [21,22], and simulations of the air-water entrainment and wave breaking [23,24]. Comprehensive reviews of the IB method applications are presented in Anderson et al. [25], Scardovelli and Zaleski [26], and Mittal and Iaccarion [27].

In the modeling of the fluid-structure interaction, the IB method is very useful to solve the problems with efficiency and ease of generation Cartesian grids [28]. Shen and Chan [29,30] developed the combined IB-VOF model to simulate the surface waves propagating over submerged bars. The studies concluded that the combined model is effective to treat the irregular rigid boundaries, giving rise to the accurate predictions of free surface evolutions and velocity features around the structures as well as the drag force. The modified IB-VOF model was suggested by Zhang et al. [31], which was capable of computing free surface transformations and their interaction with stationary or moving structures. It is noted that the difficulty of the VOF method is that complicated procedures of the interface reconstruction is required during computation [32,33]. In contrast, the level set (LS) method is a relatively simple alternative to track the front interface. It is shown that the combined scheme of LS and IB becomes popular because the method is cost effective and useful to characterize problems of the wave-structure interaction in the multiphase flows [34–36]. Recently, Lo et al. [37] integrated the LS and IB methods to study the dynamic response for, particularly, the floating body induced by the surface waves.

In the present study, a hydrodynamic model combining LS and GCIB methods [20] is developed to predict the phenomenon of nonbreaking waves passing over submerged objects. Moving interface between air and water is tracked using LS method based on the weighted essentially non-oscillatory (WENO) scheme [35,38]. The GCIB method is used to treat the interaction between fluid and immersed bodies in which a novel weighted interpolation method, accounting for the influence of arbitrary solid boundary, is contrived to accurately calculate the velocity and hence the virtual force at the image point. In addition to solving the highly deformed free surface in the process of the wave transformation at the air-water interface, the modeling gives attention to the simulations of vortex formations and

drag and lift forces produced by the submerged structures with three configurations, which are the semicircular, trapezoidal, and triangular types. For the specific shape, Young and Testik [39] observed the wave reflection coefficient and Jiang et al. [40] studied the wave loads on semicircular submerged breakwaters. However, the examination of forces acting on triangular submerged bars appear to be rare. The comparison between the three types shows that the triangular bar produces smaller drag forces (discussed in Section 3.3.3). This is important for potentially optimal design of the submerged breakwaters.

The numerical experiments are achieved via generation of the monochromatic surface waves with various periods in a wave flume. The computational effort is accordingly conducted to solve 2D Naiver-Stokes equations for the present air-water two-phase model with the interaction between the surface waves and the submerged structure. The structure of this paper is as follows. Section 2 explains the present numerical scheme and procedure of computation. Section 3 presents the numerical results of the numerical wave generations and their transformations validated with the physical experiment and the further simulations are carried out on the generated waves passing over the different types of the submerged structures. Conclusion remarks are made in Section 4.

## 2. Numerical Model

### 2.1. Governing Equation

To describe surface waves propagating in a wave flume with the immersed rigid body, the governing equations are formulated with the conservation of mass and momentum for the air-water two-layer flows with different densities and viscosities, where the interface is defined using the LS function ($\phi$) and Dirac delta function $\delta$ (explained in Section 2.2). Considering the surface tension, body force, gravity, and wave-absorbing layer, the equations are expressed as follows:

$$\nabla \cdot \mathbf{u} = 0 \tag{1}$$

$$\frac{\partial \mathbf{u}}{\partial t} + \nabla \mathbf{u}\mathbf{u} = -\frac{\nabla p}{\rho(\phi)} + \frac{\mu(\phi)}{\rho(\phi)}\nabla \cdot \left(\nabla \mathbf{u} + \nabla^{\mathrm{T}}\mathbf{u}\right) - \frac{\sigma k(\phi)\nabla\phi\delta(\phi)}{\rho(\phi)} + \mathbf{g} + \mathbf{f} - A_b\mathbf{u} \tag{2}$$

Equation (2) is the Navier-Stokes equation with the integral air and water two phase domains where the bold symbol hereinafter represents *a* vector, $\mathbf{u}$ denotes the velocity, t denotes time, $\rho$ denotes the density, $\mu$ denotes the viscosity, $\sigma$ denotes the coefficient of surface tension, *k* denotes the curvature of the air-water interface, $\mathbf{f}$ denotes the forces induced by the immersed body, $\mathbf{g}$ denotes the gravitational acceleration, p denotes the pressure, and $A_b$ denotes the absorbing coefficient which is proposed by Lin and Liu [41] and formulated as follows:

$$A_b = C_\alpha \frac{\exp\left[\left(\frac{|x - x_{xt}|}{x_{ab}}\right)^{n_c}\right] - 1}{\exp(1) - 1}, \ x_{xt} < x < x_{xt} + x_{ab} \tag{3}$$

where $x_{xt}$ and $x_{ab}$ represent the starting position and length of the absorbing region. $C_\alpha$ and $n_c$ represent the empirical damping coefficients to be determined via the numerical test where $C_\alpha$ = min {200, 0.9/$\Delta$t}, and $\Delta$t is the computational time step. With this method $C_\alpha$ ensures a positive value to prevent a negative value when the time step is large. $C_\alpha$ = 200 and $n_c$ = 10 was recommended by [41], which are also used in the prediction of motions for the floating structure in [37]. The damping region with source function $-A_b u$ is implemented to be the sponge layer in this study.

### 2.2. Level Set Method

The air-water interfacial flow is modeled using the scalar LS function to track the transient movement of the free surface. The function represents the shortest distance from the grid cell to the interface with negative value representing the air phase, positive value representing the water phase,

and zero representing the interface. The LS function is governed by the transport equation expressed as follows:

$$\frac{\partial \phi}{\partial t} + \mathbf{u} \cdot \nabla \phi = 0 \tag{4}$$

In the Navier-Stokes equation to solve the two-layer fluid flows, significant gradient of the fluid properties ρ and μ across the interface yields numerical instabilities. To eliminate these interfacial discontinuities, a thin layer ε is facilitated using a smoothed Heaviside function H(ϕ) and Dirac Delta function δ(ϕ). The transition layer of the density and viscosity distributed over the interface is shown as follows:

$$\rho(\phi) = \rho_w H(\phi) + \rho_a(1 - H(\phi)), \ \mu(\phi) = \mu_w H(\phi) + \mu_a(1 - H(\phi)) \tag{5}$$

where the subscribe a denotes the upper layer in the air and w denotes the lower layer in the water and H(ϕ) represents the smoothed Heaviside function:

$$H(\phi) = \begin{cases} 0 & \phi < -\varepsilon \\ \frac{1}{2}\left(1 + \frac{\phi}{\varepsilon} + \frac{1}{\pi}\sin\left(\frac{\pi\phi}{\varepsilon}\right)\right) & -\varepsilon \leq \phi \leq \varepsilon \\ 1 & \phi > \varepsilon \end{cases} \tag{6}$$

Typical values for ε are required between one or three times of the smallest grid size. In the present model using the uniform grid, ε is given with the value of 2Δz where Δz denotes the vertical grid size. The smoothed delta function is the derivative of the smoothed Heaviside function given by:

$$\delta(\phi) = \frac{dH}{d\phi} = \begin{cases} \frac{1}{2}\left(1 + \cos\left(\frac{\pi\phi}{\varepsilon}\right)\right)/\varepsilon & |\phi| \leq \varepsilon \\ 0 & \text{otherwise} \end{cases} \tag{7}$$

where the surface geometry is represented by the normal vector **n** and curvature $k(\phi)$ of the LS function defined as follows:

$$k(\phi) = \nabla \cdot \mathbf{n} = \nabla \cdot \left(\frac{\nabla \phi}{|\nabla \phi|}\right)_{\phi=0} \tag{8}$$

The LS equation (Equation (4)) is spatially and temporally discretized using WENO scheme [37,38] and third-order Runge-kutta scheme, respectively. As the thickness of the interface has to be uniform, each fractional time step a function is a signed distance function near the front interface. The previous LS function $\phi(\mathbf{x}, t_n)$ is updated to new $\phi(\mathbf{x}, t_{n+1})$ corresponding to the new profile of interface by solving Equation (4).

*2.3. Numerical Scheme Based on Projection Method*

The time-dependent Navier-Stokes equation is discretized using the finite difference method with the staggered arrangement of the uniform Cartesian grids. The fractional step projection method [42] is employed to decouple the velocity and pressure. The direct numerical simulation is adopted to solve the unsteady Navier-Stokes equation. To calculate the velocities of the time step from n to n+1, the convective and diffusion terms in Equation (2) are manipulated using the second-order Adams-Bashforth scheme and an implicit Crank-Nicolson method, respectively, which are second-order accuracy and ensure the numerical stability. The following predictor-corrector scheme based on the projection method is used to discretize the continuity and momentum equations given as follows:

$$\frac{\mathbf{u}^* - \mathbf{u}^{n-1}}{\Delta t} = \frac{1}{2}(3C^{n-1} - C^{n-2}) + \frac{1}{2}(D^* + D^{n-1}) - \frac{\nabla P^{n-1}}{\rho(\phi)} + \mathbf{f}_\alpha^{n-1} + \mathbf{f}^n \tag{9}$$

$$\nabla \cdot \left(\frac{\nabla p}{\rho(\phi)}\right)^n = \frac{1}{\Delta t}\nabla \mathbf{u}^* + \nabla \cdot \left(\frac{\nabla p}{\rho(\phi)}\right)^{n-1} \tag{10}$$

$$\mathbf{u}^n = \mathbf{u}^* - \Delta t\left(\frac{\nabla P^n}{\rho(\phi)} - \frac{\nabla P^{n-1}}{\rho(\phi)}\right) \tag{11}$$

where the superscript n denotes the time step and * denotes the intermediate values, $\Delta t$ represents the size of time step, $\mathbf{u}^*$ is the predicted velocities, $u^n$ and $u^{n-1}$ denote the divergence-free velocities, $P^n$ and $P^{n-1}$ related to the pressure field at time-step n and n−1, respectively, C and D represent the convective terms and diffusion terms, respectively, $\mathbf{f}_\alpha$ represents the forces including the surface tension, gravity, and wave absorbing.

## 2.4. Evaluation of Forcing Term and Ghost Cell

The velocity distribution constrained around the immersed structure is modeled using GCIB method. The forcing term $\mathbf{f}$ imposed on the Navier-Stokes equation shown in Equation (9) compensates the differences between the intermediate velocities and the resultant velocities at the ghost-cell forcing points. As the intermediate velocity is undetermined, the forcing term cannot be known explicitly. Hence, an iterative procedure is contrived to compute the intermediate velocity $\mathbf{u}^*$ and forcing term $\mathbf{f}^n$ in each time step to make the velocity at the boundary surface in agreement with the ghost-cell velocity $\mathbf{u}_g$. The forcing term is therefore obtained via rearranging Equation (9) with the implementation of $\mathbf{u}_g$ expressed as follows:

$$\mathbf{f}^n = \frac{\mathbf{u}_g - \mathbf{u}^{n-1}}{\Delta t} + \frac{1}{2}\left(3C^{n-1} - C^{n-2}\right) - \frac{1}{2}\left(D^* + D^{n-1}\right) + \frac{\nabla P^{n-1}}{\rho(\phi)} - \mathbf{f}_\alpha^{n-1} \tag{12}$$

A novel interpolation technique is proposed to calculate the velocity at the image point using the velocities of the surrounding grid points as illustrated in Figure 1. The figure sketches the ghost forcing point in the solid cell, body intercept point at the immersed boundary, and corresponding image point in the fluid cell. The three points are located at the line normal to the boundary with the intercept point at the middle. It is observed that an image point is usually projected inside the rectangular fluid cell enclosed by the four fluid points as shown in Figure 1a. Additionally, the image point is probable to locate inside the cell in presence of the immersed boundary. As illustrated in Figure 1b the image point is enclosed by three fluid points and the body intercept point. The image velocity $\mathbf{u}_p$ is therefore interpolated using the velocities at the three fluid points and at the body intercept point, which forms the quadrilateral stencil marked as dash line in Figure 1b. A particular case is given in Figure 1c because the image point is surrounded by two fluid points, one solid point, and the intercept point. To solve this problem, the stencil is extended to include the nearest fluid cell and hence the $\mathbf{u}_p$ is interpolated by the all fluid points on the stencil and the body intercept point.

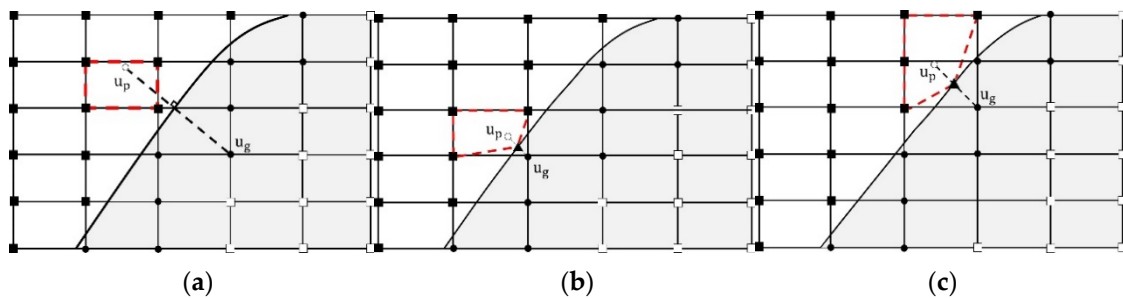

(a)　　　　　　　　　　　　　(b)　　　　　　　　　　　　　(c)

**Figure 1.** Uniform Cartesian grids around the immersed boundary with three types of the image points inside the rectangular cell (**a**) in the fluid without solid point, (**b**) crossed by the boundary with one solid point, and (**c**) crossed by the boundary with two solid points. The points on the stencil marked by the dashed line are used to compute the velocity $u_g$ at the ghost cells. (● Ghost-cell forcing points; ○ Image points; ■ Fluid points; □ Solid points; ▲ Body intercept point).

The image velocity is obtained using the weighting function:

$$\mathbf{u}_p = \sum_{i=1}^{ns} w_i^k \mathbf{u}_i \tag{13}$$

where $w_i$ is the weight of grid point, subscript i indicates the number represented the fluid and body intercept points used to interpolate $\mathbf{u}_p$ and ns represents the total number of the quadrilateral stencil, superscript k marks the kth image points in the computational domain, and w represents the weighting function. The value of weighting function ensures the consistent condition of the total velocity source from the surrounding points to the image point. The weighting summation of kth marker point satisfies:

$$\sum_{i=1}^{ns} w_i^k = 1 \tag{14}$$

The weighting function for the image node to smooth a quantity from the kth marker point is given by:

$$w_i^k = \frac{\widetilde{w}_i^k}{\sum_{i=1}^{ns} \widetilde{w}_i^k} \tag{15}$$

The weight is obtained using the reciprocal of distances between the image point and the interpolated points represented as:

$$\widetilde{w}_i^k = 1/\left(r_{pi} + \alpha\right) \tag{16}$$

where $\alpha$ represents the regularization parameter with a constant value of $10^{-6}$ and $r_{pi} = \sqrt{\left(x_p - x_i\right)^2 + \left(z_p - z_i\right)^2}$ denotes the distance to the position of the image point $\left(x_p, z_p\right)$. If the image and interpolated point is identically located or the distance between two points is very small, Equation (16) approaches to infinite without adding $\alpha$. The values at ghost-cell forcing points are computed using the present interpolation method with the capability of minimizing the numerical instability.

After calculation of the velocity at the image point, the velocity at ghost cell point is extrapolated through the boundary condition and hence:

$$\mathbf{u}_g = 2\mathbf{u}_\Gamma - \mathbf{u}_p \tag{17}$$

where $\mathbf{u}_\Gamma$ denotes the velocity at the boundary.

For the engineering design of breakwaters, it is important to evaluate the wave force acting on the structure. This is calculated directly using the pressure and the viscous shear stress on the wall and integrating along with the surface $\Omega$ of the solid body:

$$\mathbf{F} = \int_\Omega \left(-\mathbf{n}p + \mathbf{n}\cdot\boldsymbol{\tau}\right)d\Omega \tag{18}$$

where $\boldsymbol{\tau}$ represents the shear stress on the wall. As the Navier-Stokes equation formulated in Equation (2) includes the gravity force the pressure indicates the summation of the hydrostatic and dynamic pressure.

*2.5. The Numerical Procedure*

The computational procedure for the present numerical scheme is described as follows:

1.　Equation (9) is solved to obtain the intermediate velocity $\mathbf{u}^*$ without consideration of the forcing term $\mathbf{f}^n$. However, at this moment $\mathbf{u}^*$ does not satisfy the boundary condition at the rigid body surface.

2. The velocity at the forcing point is computed using the interpolation method and hence the forcing term $\mathbf{f}^n$ in Equation (12) is obtained.
3. Equation (9) is recalculated to provide a new $\mathbf{u}^*$. As the forcing term $\mathbf{f}^n$ and intermediate velocity $\mathbf{u}^*$ are implicit variables, an iterative scheme between steps 2 and 3 is used to ensure the convergence of both $\mathbf{f}^n$ and $\mathbf{u}^*$.
4. Pressure is computed via the Pressure Poisson equation of Equation (10).
5. The velocity is corrected using Equation (11).
6. In the successive time step, the velocity, forcing function, and pressure calculated from the previous time step are employed to be the initial conditions. The above explained procedure is iterated until the required time step is completed.

A code was written to conduct the numerical modeling using a personal computer with the CPU of I5 10400. The present model took approximately 10 min to run a relatively complex case of wave passing over the submerged body with a specific wave height and period. In contrast, as given in Section 3.2.1 for the model validation, the a commercially available CFD package, Flow-3D, using the renormalized group (RNG)-turbulent flow model took approximately 12.8 min. This reveals the efficiency of the present numerical scheme with the combined LS and GCIB methods.

## 3. Results and Discussion

### 3.1. Monochromatic Wave Generations

The present model was firstly validated via the generation of progressive waves in the rectangular wave flume with a constant depth to compare with the theoretical solutions [43]. The domain size of the flume is 30 m in length, 0.4 m in water depth, and 0.2 m in height for the upper air layer as the schematic diagram shown in Figure 2. The numerical flume is consistent with the physical wave flume and layout of Luth et al. [7]. In the modeling, the horizontal and vertical grid sizes are $\Delta x = 0.02$ m and $\Delta z = 0.005$ m, respectively, with total nodes of 180,000 and time step $\Delta t = 0.002$ s. Numerical waves are simulated using the relaxation zone method [44] with the analytically small amplitude waves fed into the inlet boundary. This method is in advantage of adopting the momentum equations with the source function to generate arbitrary waves. The waves are produced in a relaxation zone with the typical range of one wavelength. At the outlet of the flume, the thick sponge, two to three times of the wavelength, is arranged to dissipate the incoming waves and hence reduce the wave reflection.

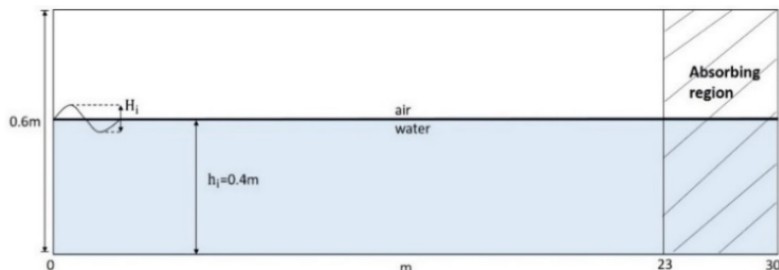

**Figure 2.** Sketch of the computational domain for the linear wave generation.

The linear wave is generated in still water with the incident wave height $H_i = 0.02$ m and period $T = 2$ s. The wavelength L is approximately 3.69 m calculated using the linear dispersion relationship in the intermediate water depth. As the steepness $kA_0 = 0.017$ is significantly smaller than the nonlinear wave criteria of 0.1, where k represents wavenumber and $A_0$ represents the incident wave amplitude, the nonlinear effect is not considerable and the waveform is maintained uniformly with respect to the downstream distance [45]. The numerical results of the wave profiles are shown in Figure 3 with the dimensionless form of $\eta/A_0$ where $\eta$ indicates the surface elevation observed at two wave gauges with the downstream distances of x = 19 and 21 m, respectively. The wave

amplitudes show regular and consistent at the two stations with time. The numerical wave profiles are in very good agreement with the analytical solutions, particularly with the dimensionless time $t/T$ larger than 10. Small discrepancies are observed with $t/T$ smaller than 10, which appears in the development stage with weak instability. The root mean square (RMS) errors between the numerical and analytical results during several periods are 0.1% and 0.12 % at the two gauge stations of $x = 19$ m and $x = 21$ m, respectively.

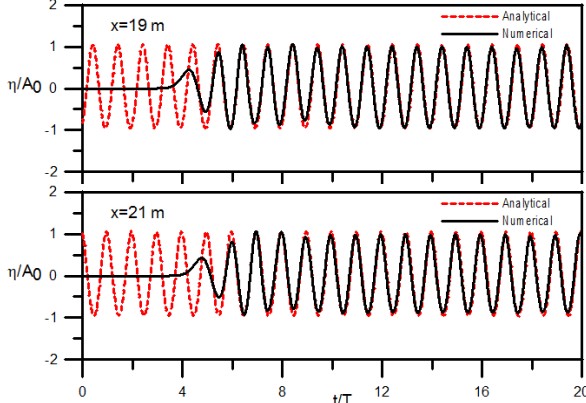

**Figure 3.** Simulation results of the surface elevation for linear progressive waves at the two gauge stations of $x = 19$ m (**upper**) and $x = 21$ m (**lower**) with the incident wave amplitude $H_i = 0.02$ m and period $T = 2$ s.

The test of grid independence is conducted using the coarse grids with the mesh size $\Delta x = 0.04$ m and $\Delta z = 0.01$ m. The number of the nodes is reduced to 45,000. Figure 4 presents the very good agreement between the simulation results with the two different mesh sizes at the two stations, indicating the grid convergence.

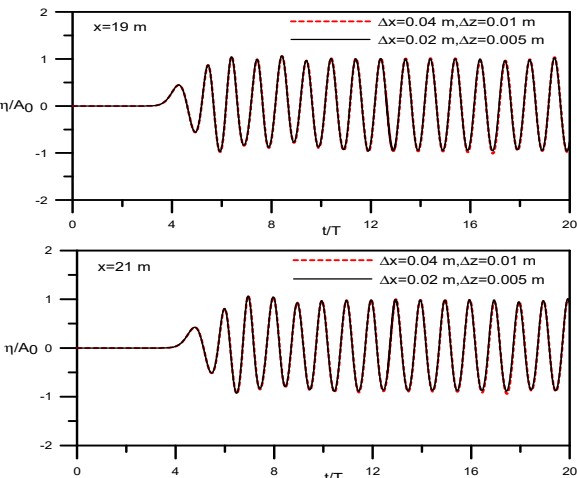

**Figure 4.** Simulation results for grid independent test using two different mesh sizes at the two gauge stations of $x = 19$ m (**upper**) and $x = 21$ m (**lower**). The condition of wave generation is identical to Figure 3.

### 3.2. Wave Transformation Over a Trapezoidal Submerged Breakwater

Wave transformation over a submerged bar involves in the classical problem of nonlinear wave shoaling to produce the higher harmonics and redistribution of the wave energy between the primary wave and high frequency waves. Beji and Battjes and Luth et al. [6,7], summarized by Dingemans [46], conducted the well-known laboratory experiment to examine the waves transformed over a trapezoidal submerged bar. The experiments become standards to validate the numerical study of the wave

transformation caused by the submerged breakwater. To verify the present two-phase flow model, the observation of Luth et al. [7] is employed to compare with the numerical results. More studies are carried out to observe wave transformations with different wavelengths.

### 3.2.1. Validation

The computational domain and layout of the submerged bar is displayed in Figure 5. The incident waves are numerically generated in the relaxation zone with the wave height and period of $H_i = 0.02$ m and $T = 2.02$ s. Free surface elevations are measured at nine downstream wave gauges denoted as Station P1 to Station P9 as sketched in Figure 5. The flume layout, condition of the wave generation, and measurement locations are in agreement with the experiment of Luth et al. [7].

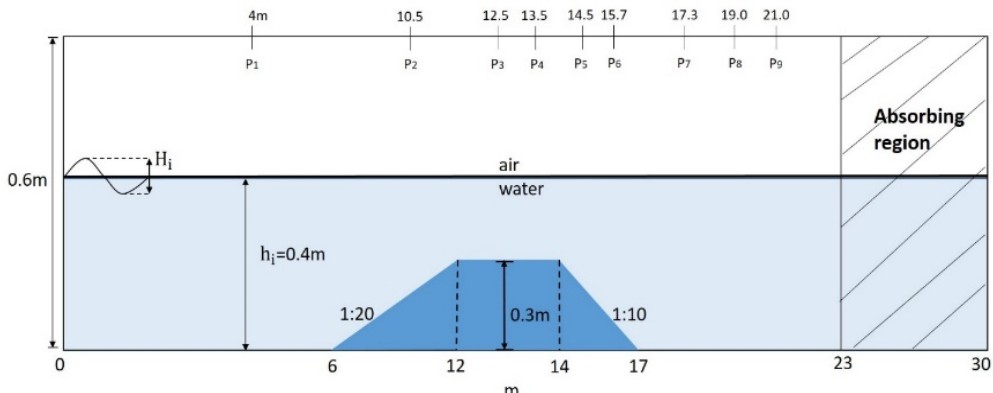

**Figure 5.** Sketch of the computational domain for the linear waves passing over the trapezoid breakwater.

Figure 6 shows the prediction of the wave evolution at the nine stations in comparison with the simulation of the Flow-3D and the experimental measurements. The Flow-3D computation is based on the renormalized group (RNG)-turbulent flow model. Both models successfully simulate the wave transformation over the obstacle and demonstrate good agreement with the experiment [7] in terms of the development of high frequency waves. Better agreement is given by the present predictions at each station from Station P1 to P9, particularly for the highly deformed surface after the waves passing the submerged object and arriving at Station P8 and P9 as shown in Figure 6. The discrepancies between the simulations and the experimental observation are quantified using the RMS error analysis.

The two largest amplitude differences are given at Station P4 over the trapezoidal bar and Station P9 far away from the bar with the errors of 5.12% and 4.42% for the present model, and 6.13% and 5.63% for the Flow-3D. The corresponding phase errors are of 0.92% and 1.03% for the present model and 3.31% and 2.66% for the Flow-3D. The results indicate that the present numerical scheme is capable of accurately predicting the surface distortion with the rapid growth of the higher harmonic waves at the air-water interface with high resolution.

### 3.2.2. Wave Transformation with Three Wavelengths

To understand the influence of wavelengths on the transformation, the incident wave height $H_i = 0.02$ associated with three different wave periods with $T = 1.01$ s, 2.02 s (the validation case in Section 3.2.1), and 3.03 s are produced with the computational domain and embedded body identical to Figure 5. As explained in Section 3.1, for these waves with very small steepness, they propagate monochromatically with respect to the downstream distance to the stage experiencing the external force driven by the rigid bar in the water.

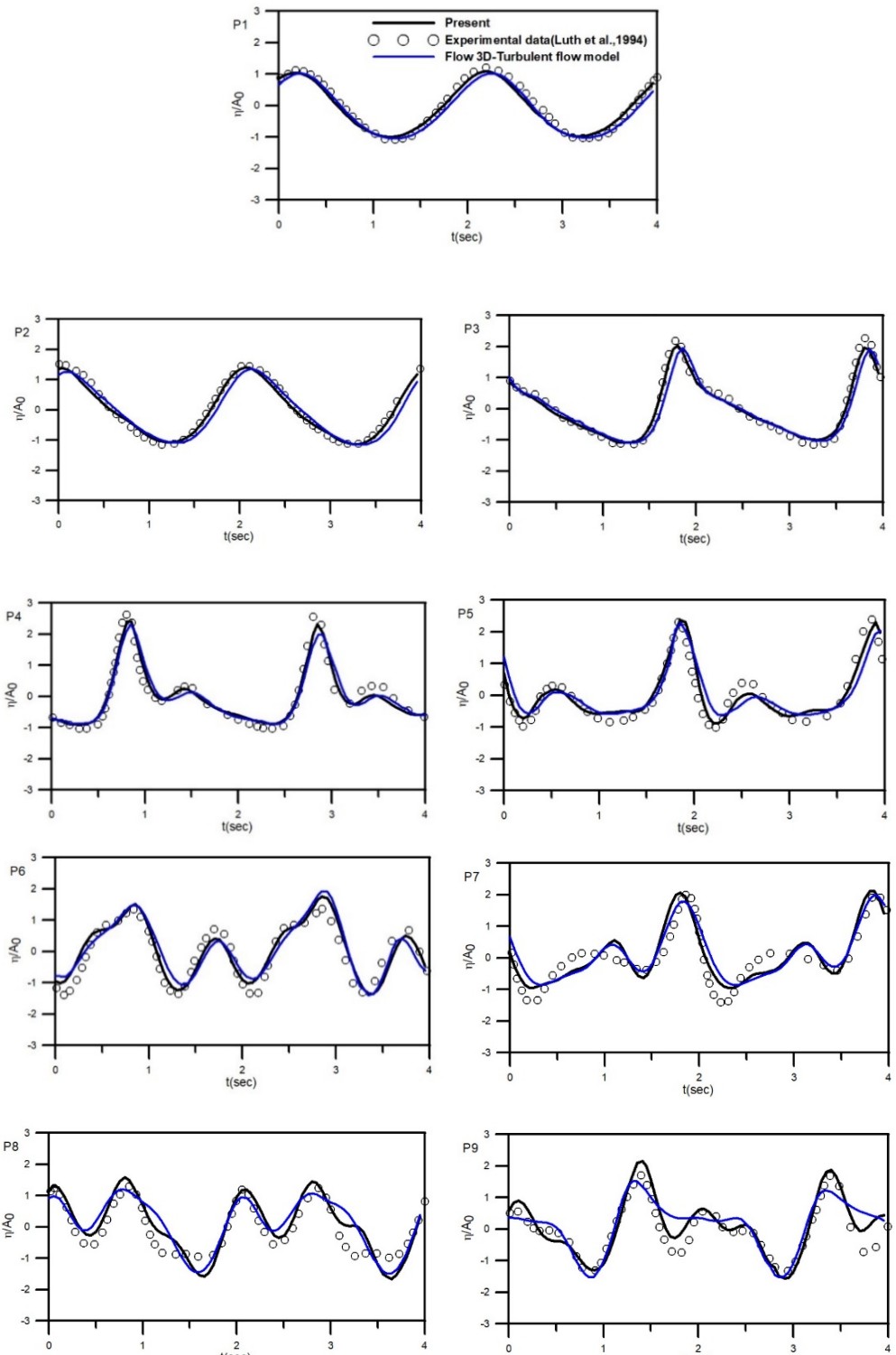

**Figure 6.** Simulation of surface elevations compared with the experiment at nine wave gauges from Station P1 to P9 for wave passing over the trapezoidal bar with the incident wave height $H_i = 0.02$ m and period T = 2.02 s.

The surface undulation due to wave decomposition with the dimensionless time series scaled by the wave period were analyzed using the fast Fourier transform to observe the higher harmonics in the amplitude spectra as demonstrated in Figure 7 with T = 2.02 s. At Station P2 over the upward slope of the trapezoidal dike, the shoaling effect has caused the moderate growth of the second harmonic on

the amplitude spectrum as shown in Figure 7b associated with the asymmetry of the primary wave profile slightly titled backwards and positive skewness with the larger amplitude of the crest than the trough. The definition of wave asymmetry and skewness was explained by Bananin et al. [47]. As the waves continuously deformed at Station P3 over the crest surface of the bar, substantial increase of skewness and asymmetry related to the growth of the third and fourth harmonics are observed due to the significant wave shoaling effect. When the waves travel further downstream with the crest at Station P4, Figure 7d demonstrates that small and high frequency waves appear at the trailing edge of the dominant waves [13]. Meanwhile, it is noted the considerable decrease of the amplitude at the dominant frequency and increase of the high frequency waves. For the Stations P5 and P6 over the downward slope, Figure 7f reveals that the wave profiles become more sophisticated and the spectra show comparable amplitudes between the primary and secondary harmonics. With the waves moving away the submerged structure and transmitted to the deeper water at Stations P7, P8, and P9, the surface oscillation appears unpredictable in spite of regular recurrence of the profile in a wave period. The spectrum in Figure 7g shows the higher amplitude of the second harmonic than the primary component, indicating that the nonlinear mechanism primarily varies the originally sinusoidal waves.

It is interesting to know the wave deformations influenced by the wave periods, that is, the wave wavelengths. Figure 8 demonstrates the prediction results for the incident wave period T = 1.01 s and wavelength L = 1.46 m. For this relatively short wave, the wave amplitudes behave slightly unevenly with absence of the significant shoaling effect when the waves move over the top surface of the breakwater as exhibited in Figure 8c,d. The spectra show the growth of amplitudes at the primary frequency and weak development of the second harmonics. However, after the waves pass the structure and propagate into the deeper water region, Figure 8h,i shows the disappearance of the second harmonics. The viscous effect dissipates the small and higher harmonic waves and the waves return to the sinusoidal profiles.

The transformation arising from the incident wave period T = 3.03 s and wavelength L = 5.77 m is demonstrated in Figure 9. With this longer wavelength passing over the crest of the submerged structure at Stations P3 and P4, the asymmetry and skewness of wave profiles are more significant than the condition of T = 2.02 s shown in Figure 7, indicating a relatively strong wave shoaling due to the increase of the ratio of $L/qh_i = 57.7$ where $h_i$ denotes the still water depth, q denotes the ratio of the still water to the shallow water depth, and $qh_i$ denotes the shallow water depth above the crest of the structure. Figure 9e,f reveals that the effect continuously influences the waveform with rapid growth of high frequency waves at Stations P5 and P6. Besides, with wave transmitted to Stations P7 to P9 at the deeper water region, Figure 9g,f shows that the formation of the distinctly irregular waves is associated with the energy distributed from primary waves to the high frequency waves of the second to fourth harmonics in the wave spectra.

*3.3. Progressive Waves Over Three Types of Submerged Breakwaters*

The monochromatic waves passing over submerged structures are further modeled with three different configurations, which are semicircle, trapezoid, and triangle, to study the velocity fields and hydrodynamic forces acting on the bodies. The semicircular submerged breakwater is interested in contributing a stable structure under the wave action [39]. All three types of the structures have equal area of 1 m$^2$ with the height of 0.8 m. Figure 10 shows the sketch of the computational domain with the flume of 60 m in length, 1 m in water depth, and 0.5 m in height for the air. Both sides of the flume were allocated with absorbing regions to prevent wave reflection. The uniform grid sizes used for the numerical computation are $\Delta x = 0.1$ m and $\Delta z = 0.05$ m, respectively. The linear waves are generated at x = −L/2 m with the incident wave height $H_i = 0.2$ m and periods T = 3 and 4 s corresponding to the wavelength L = 8.69 and 12 m. Three wave gauges are located at the downstream distance of x = 18 m in front of the obstacle, x = 20 m over the top surface at the central line of the structure, and x = 22 m over the backward slope to observe the behavior of the surface wave decomposition.

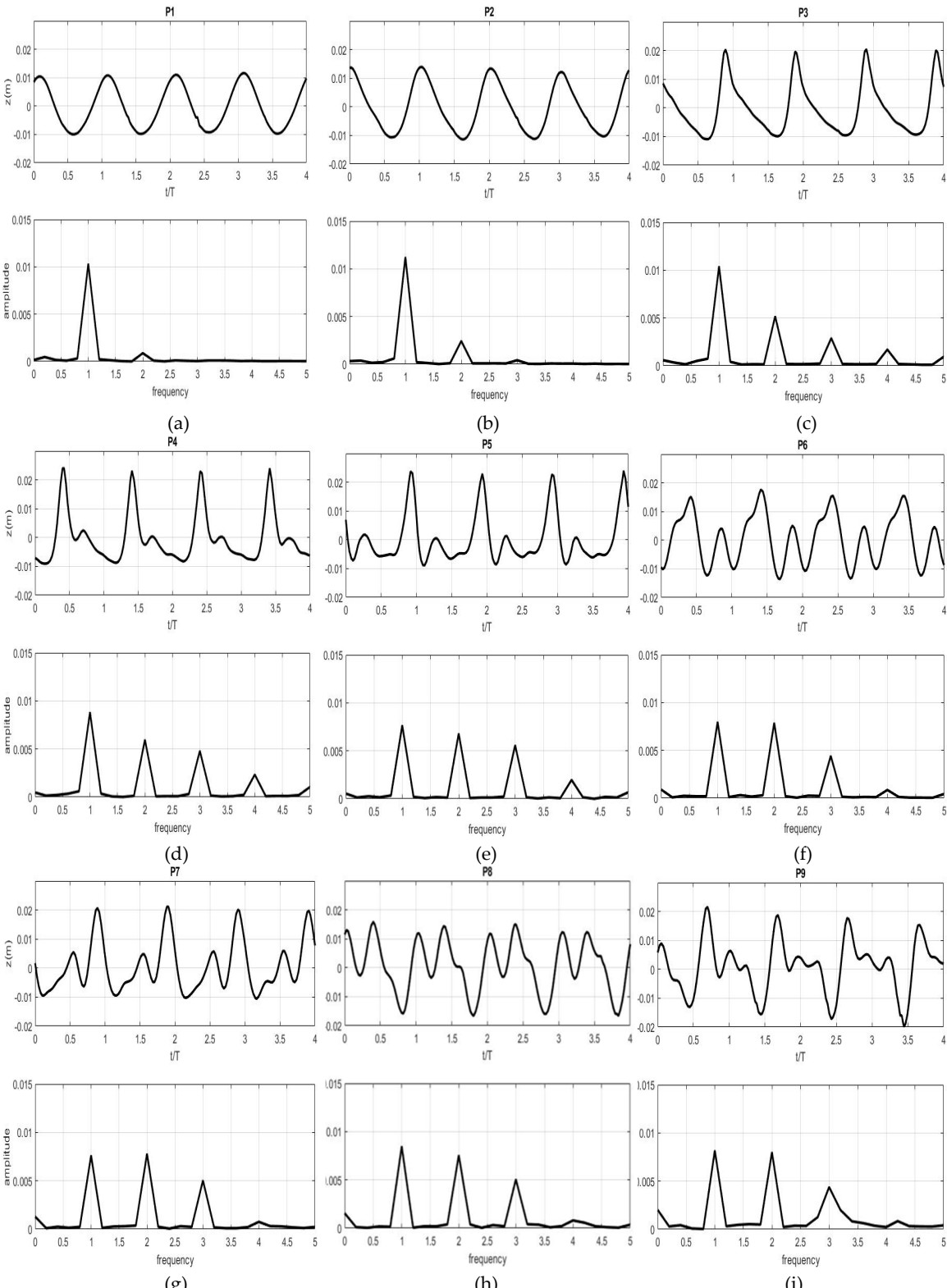

**Figure 7.** Simulation of surface elevations (upper) and amplitude spectra (lower) at nine downstream positions with the incident wave height $H_i$ = 0.02 m and period T = 2.02 s passing over the trapezoidal submerged structure. (**a**) Station P1; (**b**) Station P2; (**c**) Station P3; (**d**) Station P4; (**e**) Station P5; (**f**) Station P6; (**g**) Station P7; (**h**) Station P8; (**i**) Station P9.

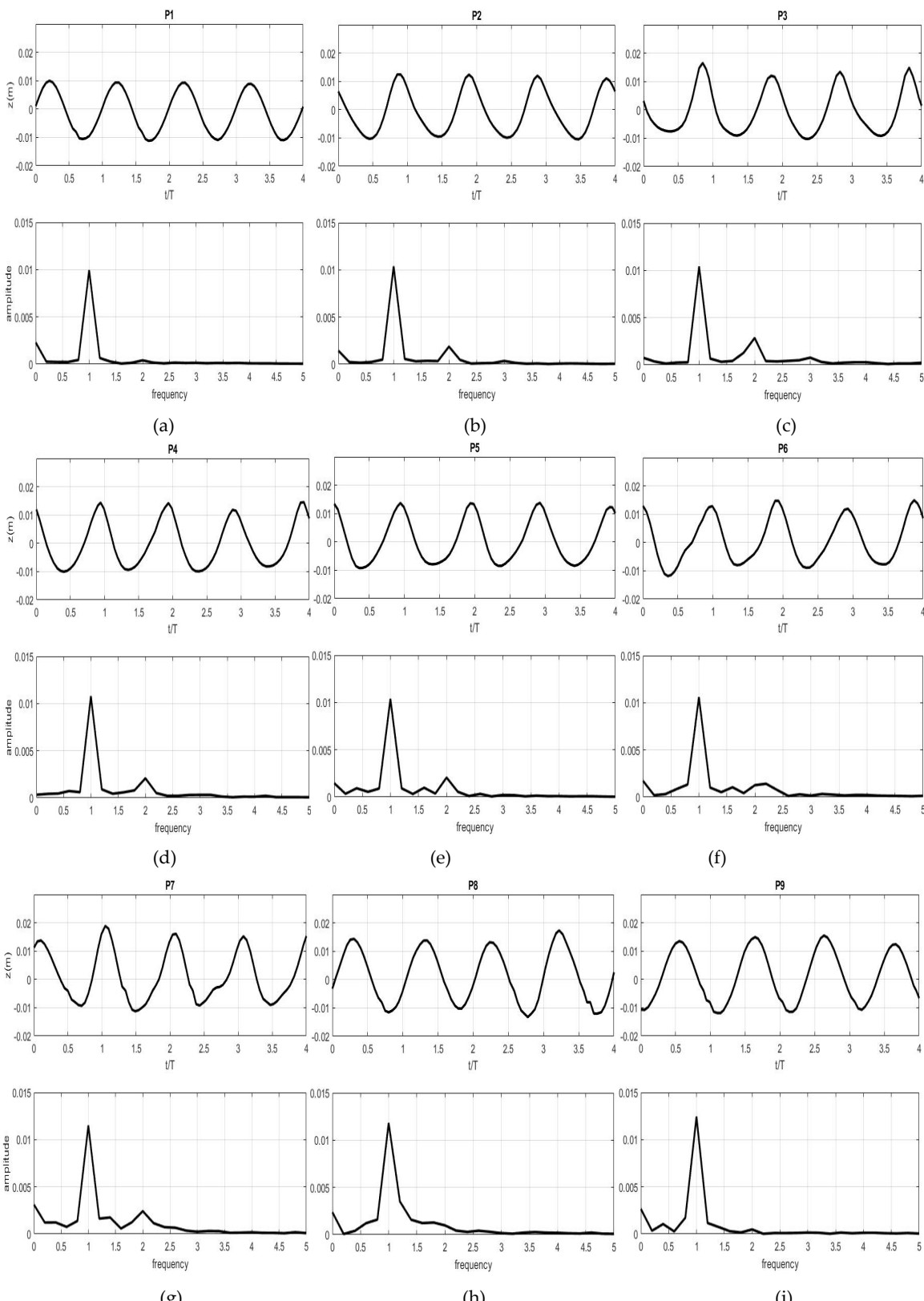

**Figure 8.** Simulation of surface elevations (upper) and amplitude spectra (lower) at nine positions with the incident wave height $H_i$ = 0.02 m and period T = 1.01 s passing over the trapezoidal submerged structure. Explanations of (**a**–**i**) are given in the caption of Figure 7.

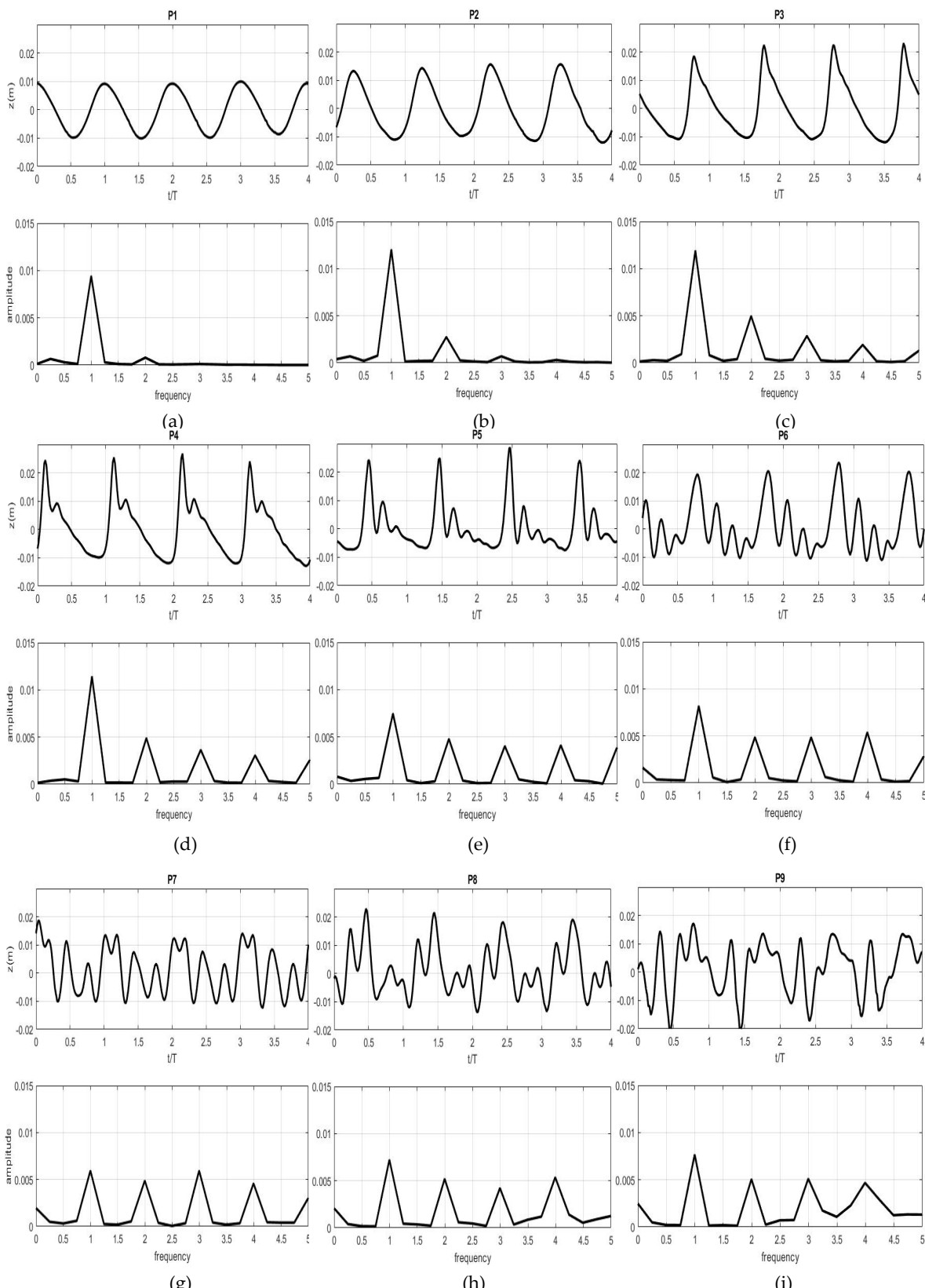

**Figure 9.** Simulation of surface elevations (upper) and amplitude spectra (lower) at nine positions with the incident wave height $H_i$ = 0.02 m and period T = 3.03 s passing over the trapezoidal submerged structure. Explanations of (**a–i**) are given in the caption of Figure 7.

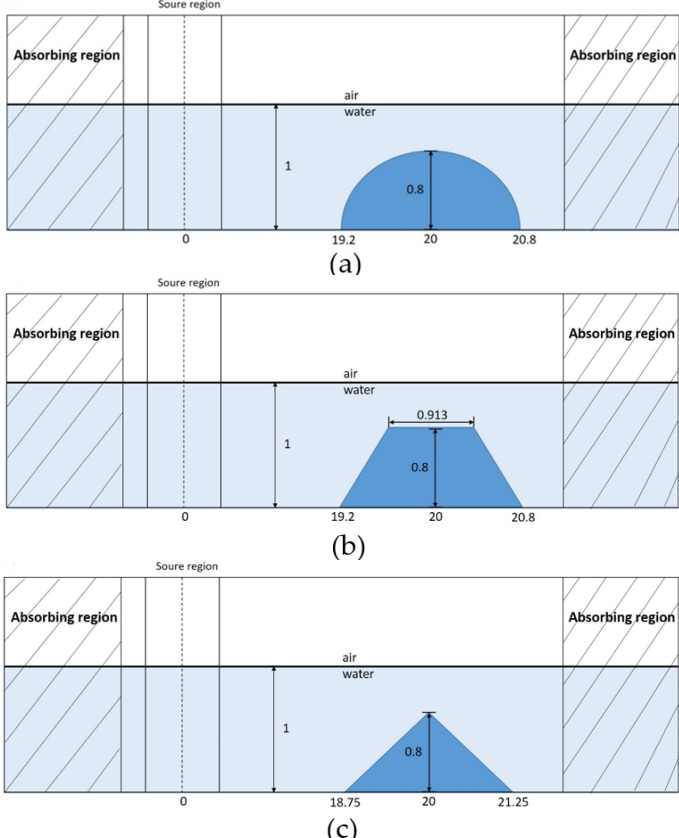

**Figure 10.** Sketch of the computational domain; for the submerged structures, (**a**) semicircle, (**b**) trapezoidal, and (**c**) triangular with the cross area of 1 m² and height of 0.8 m.

### 3.3.1. Wave Transformations

Figure 11 shows the predictions of the surface elevations with the incident wave period T = 3 s. Analogous to the wave profiles given in Figure 6, the wave-structure interaction causes the deformation of the waveform with the skewness and asymmetry considerably titled backwards at x = 18 and 20 m as given in Figure 11a,b. In addition, the wave shoaling significantly increases the amplitude with the wave over the top of the structure at x = 20 m. At the further downstream distance of x = 22 m, the wave profiles become slightly tilted forwards. It appears that the three configurations of the submerged obstacles produce insignificant discrepancies between the wave profiles. However, the high frequency waves develop at the trail edge of the primary wave for the triangular shape as demonstrated in Figure 11c. The numerical results of the wave deformation with the incident wave period T = 4 s are displayed in Figure 12. With the longer wavelength, the deformed wave profiles are consistent with the three types of the submerged structures. The waveforms with significant steepness are noted for the three configurations in Figure 12c, indicating that the formation of highly nonlinear waves with transferring the wave energy from the primary component to high frequency waves.

### 3.3.2. Velocity Patterns Around the Objects

The surface elevations and patterns of the instantaneous velocities in the procedure of the waves traveling over the semicircle bar are demonstrated for three different times in Figure 13 with the incident wave period T = 3 s. In the stage shown in Figure 13a with the wave trough over slightly upstream of the top surface, the presence of the obstacle causes the drag force to retard the nearby water flow and hence produces the significant gradient in the boundary layer, beginning approximately at x = 19 m in front of the semicircle. The increase of the velocity gradient along with the solid surface downstream finally makes the flow separation at the immersed surface beneath the wave trough. For the wave

crest propagating over the solid body slightly downstream of the top surface shown in Figure 13b, the velocity profiles with the significant curvatures move forwards and are located at x = 19.6 m on the surface of the semicircle. It is observed that a vortex is generated and rotates counterclockwise upstream of the object. In contrast, a secondary vortex is formed close to the left-toe of the obstacle and rotates clockwise. A moderate velocity gradient occurs at the leeward side and under the crest of the wave. Following the subsequent state with wave moving approximately a diameter away from the semicircle, the upstream vortices disappear. However, a small vortex is formed at the backward side of the top surface as depicted in Figure 13c. This circumstance with the wave passing away from the submerged body is analogous to the velocity patterns given by Jiang et al. [40].

The flow patterns around the obstacle are compared between three types of submerged structures as shown in Figure 14 with the generation of the incident wave period T = 3 s at t = 40 s. The temporal condition indicates the wave crest just over the top surface of the embedded objectives. For the three shapes, the velocity profiles with significant gradient are observed at approximately the equivalent locations occurring over the windward slopes. Figure 14a shows that the velocity distribution around the semicircle is in analogous to the condition presented in Figure 13b, however, with the increase of the secondary vortex in front of the left-toe of the obstacle. Different from the semicircle, a vortex appears upstream of the trapezoidal body as well as another vortex behind the trapezoidal structure. This circumstance is not observed for the other two solid objectives, which suggests that the trapezoidal shape gives rise to a larger form drag with the wave generated water flow over the immerged body. There is only a small vortex formed backward slope of the triangular bar, implying the relatively small drag force. The substantially different distribution of the velocities due to the shape of structures produced a significant effect on the hydrodynamic force when the wave is acting on the structure. This phenomenon is further discussed in Section 3.3.3.

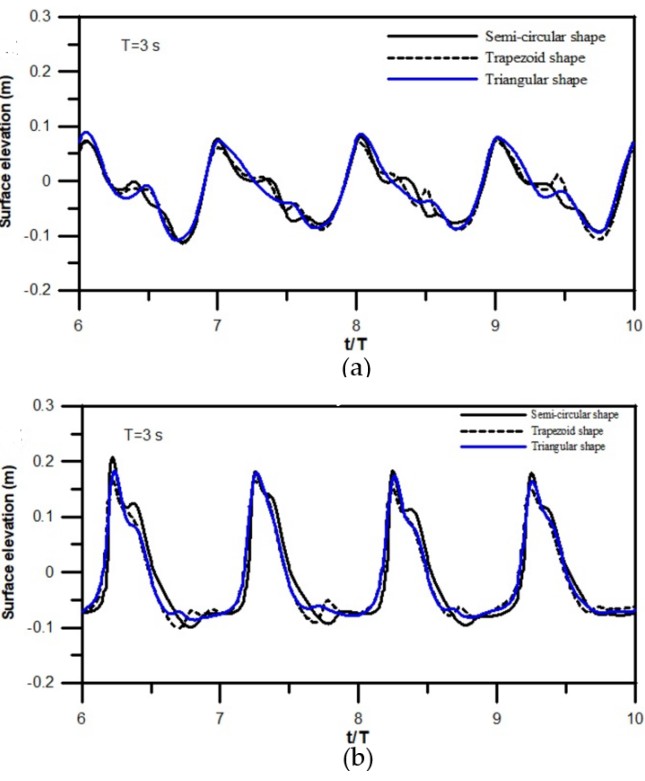

**Figure 11.** *Cont.*

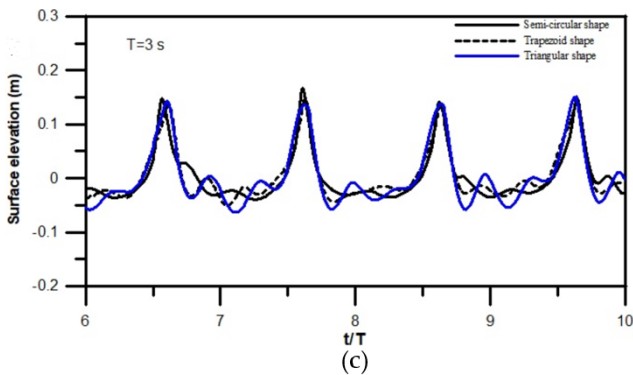

**Figure 11.** Time history of surface elevation with the incident wave height $H_i$ = 0.2 m and period T = 3 s at three wave gauges for the three types of breakwaters at the downstream distance (**a**) x = 18 m; (**b**) x = 20 m; (**c**) x = 22 m.

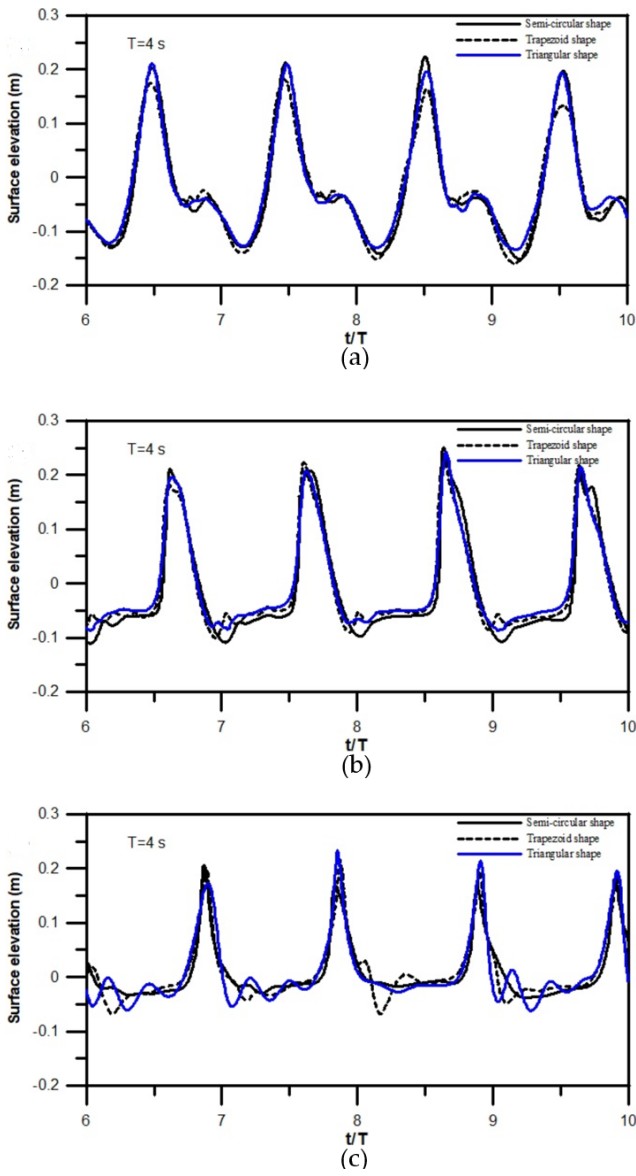

**Figure 12.** Time history of surface elevation with the incident wave height $H_i$ = 0.2 m and wave period T = 4 s at three wave gauges for the three types of breakwaters at the downstream distance (**a**) x = 18 m; (**b**) x = 20 m; (**c**) x = 22 m.

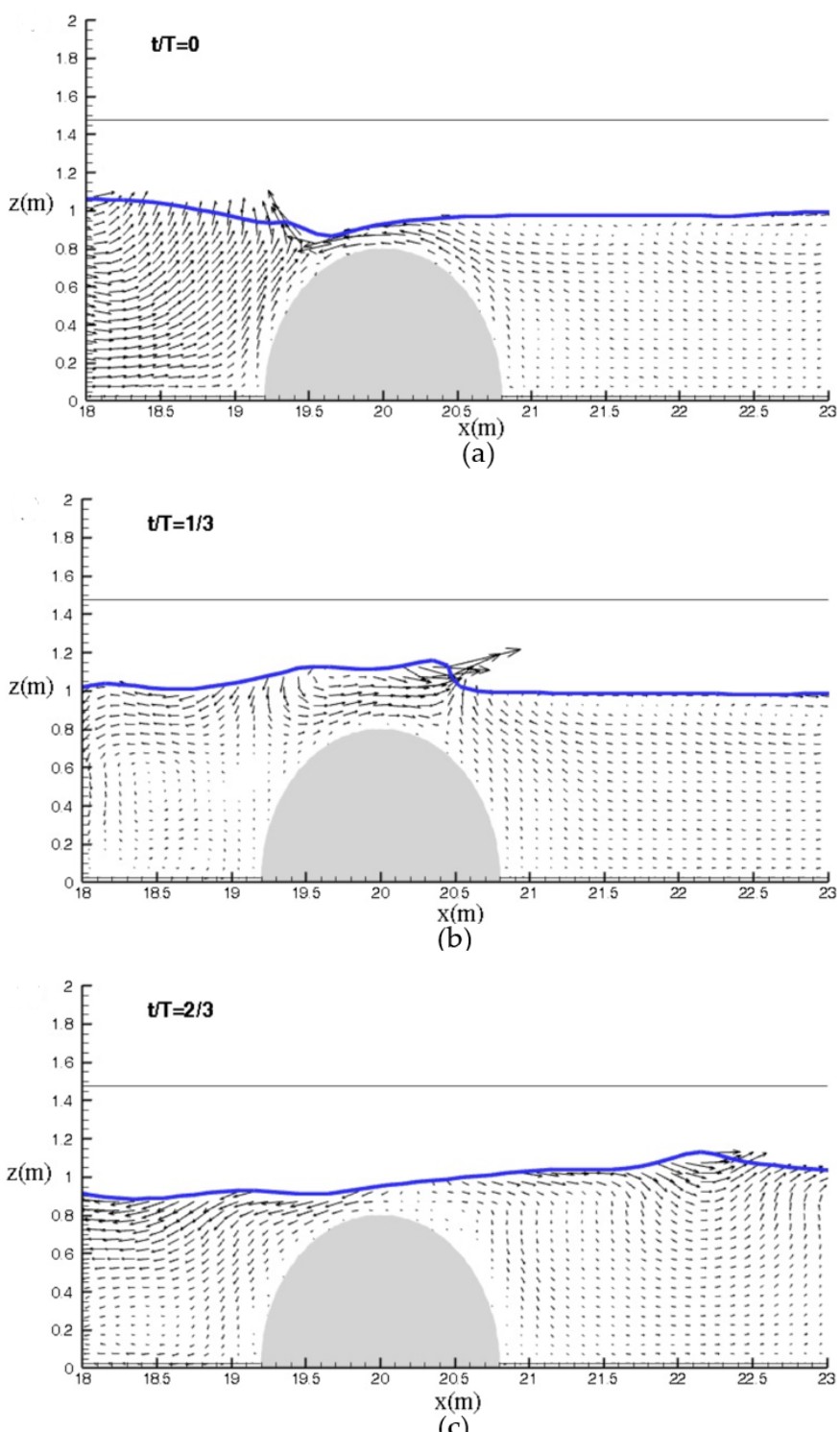

**Figure 13.** Instantaneously velocity fields for the monochromatic wave with $H_i = 0.2$ m and $T = 3$ s passing over the semicircular submerged structure for (**a**) $t/T = 0$, (**b**) $t/T = 1/3$, and (**c**) $t/T = 2/3$.

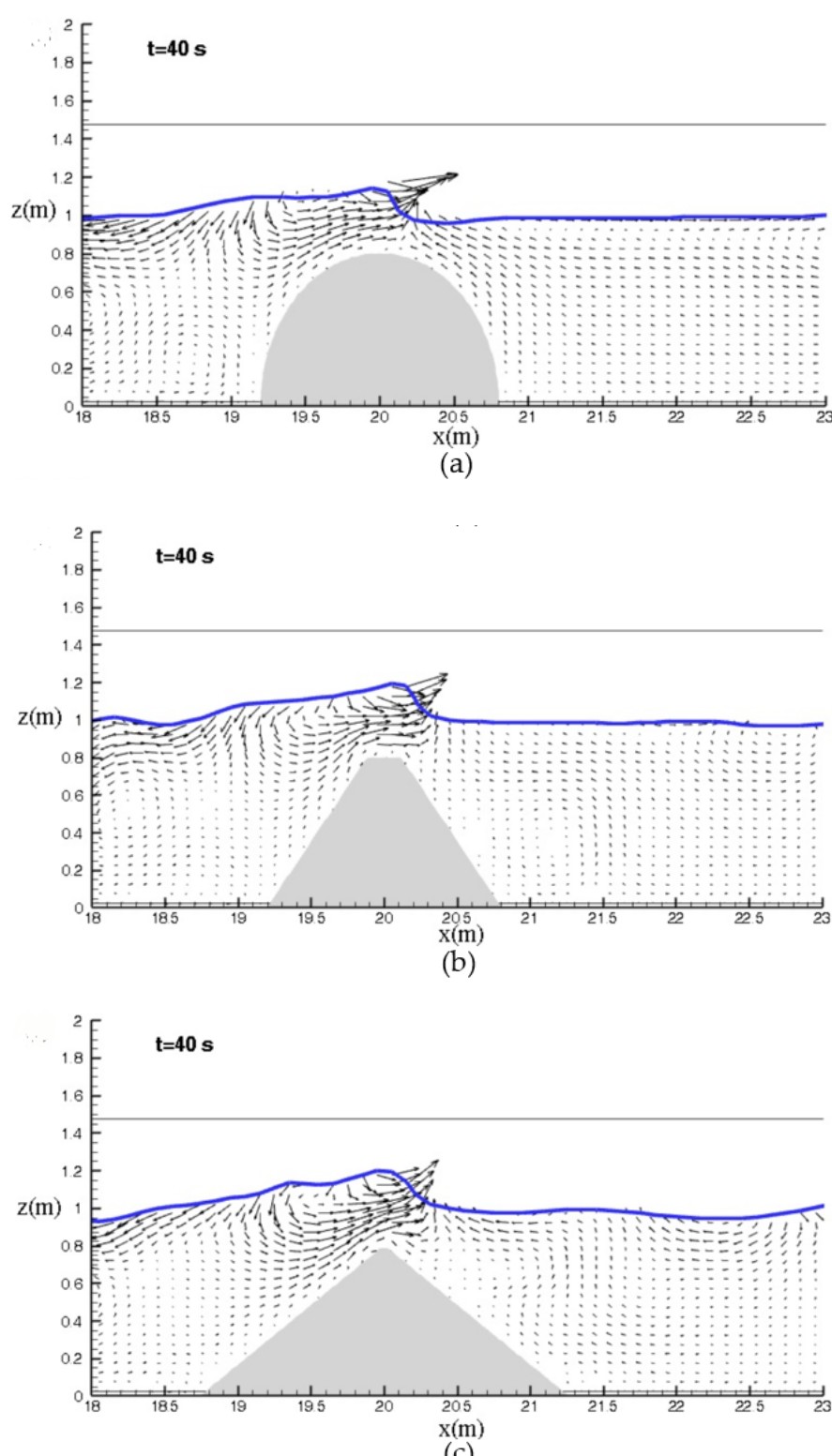

**Figure 14.** Instantaneously velocity fields at $t = 40$ s for the monochromatic wave with $H_i = 0.2$ m and $T = 3$ s passing over three types of the submerged structures of (**a**) semicircle, (**b**) trapezoid, (**c**) triangle.

### 3.3.3. Simulations of Drag and Lift Forces

Determination of the drag and lift forces of surface waves acting on marine structures is important for the practical design of breakwaters. In the present numerical solutions, the forces are computed

using Equation (18). The drag force here refers to the combination of the form drag and friction drag induced by pressure and shear stresses. A test was conducted to make a comparison between the simulations of the present model and Flow-3D. With the incident wave height of $H_i = 0.2$ m wave period of $T = 3$ s passing over the three types of submerged structures, the surface oscillation giving rise to the periodic water flow and hence the periodic variations of the drag forces with respect to time as shown in Figure 15. The maximum drag force occurs with the wave crest over the central line of the objects. Good agreement between the two simulations is exhibited although the present model produces larger force at the wave crests before $t = 20$ s and smaller force at the wave troughs after $t = 20$ s. The RMS errors of the amplitudes between the two model results give 6.45%, 7.56%, and 5.84% for the semicircular, trapezoidal, triangular types with the corresponding phase errors of 2.8%, 3.5%, and 2.4%, respectively.

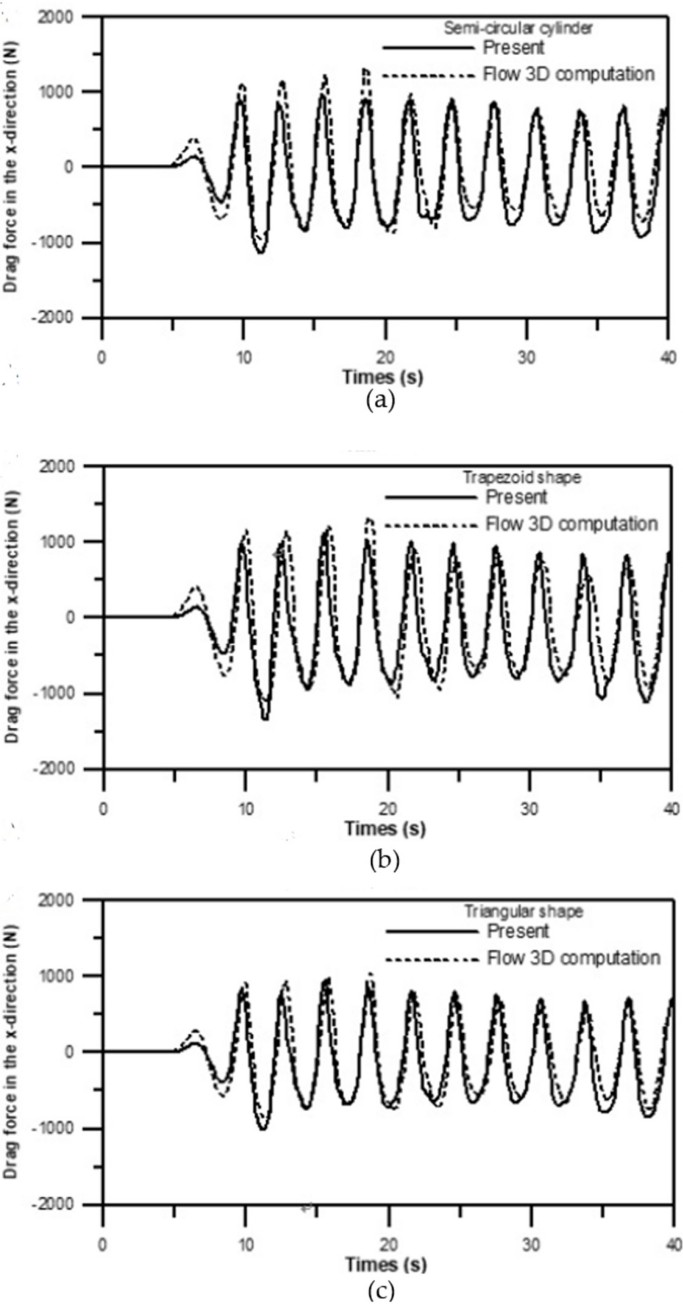

**Figure 15.** Periodic variation of the drag forces for the progressive waves with $H_i = 0.2$ m and $T = 3$ s over the breakwaters for the type of (**a**) semicircle, (**b**) trapezoid, (**c**) triangular.

The comparisons of the drag and lift forces for the three types of the structures are represented in Figure 16 with the incident wave period T = 3 s. Huang and Dong [14] modeled the form and friction drag for the solitary wave traveling over the rectangular dyke. They observed that the friction drag was only approximately 3% of the form drag, which was negligible. Hence, the drag formed in the present circumstance is mainly due to the shapes of the submerged bars. Figure 16a demonstrates that the trapezoidal object causes the largest drag while the triangular structure contributes the smallest drag force. In contrast to the velocity patterns given in Figure 14, it is likely that the drag force is highly correlated to the formation of the vortex around the obstacles. For example, as presented in Figure 14b there is a significant vortex located on the downstream of the triangular structure and hence generates the larger drag force. Figure 16b exhibits the variations of lift forces with time. Analogous to the formation of the drag force, the triangular shape also gives the smallest lift force. In contrast, the semicircular structure shows the considerable increase of lift force. This is reasonable because the semicircle is more likely to be an aerodynamic shape.

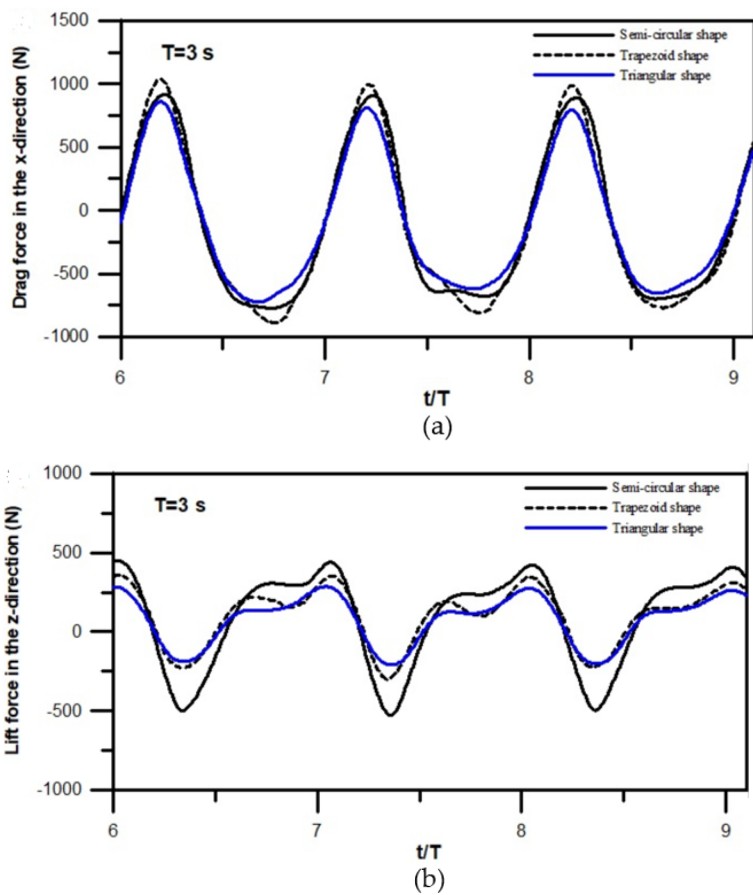

**Figure 16.** Simulation of the (**a**) drag and (**b**) lift forces for the different configurations of the submerged structures with the incident wave height $H_i = 0.2$ m and T = 3 s.

The drag and lift forces with the incident wave period T = 4 s are demonstrated in Figure 17. The drag force significantly increases approximately 50% for the longer wavelength with L = 12 m when compared with the shorter wavelength with L = 8.96 m as depicted in Figures 16a and 17a. This is because a long wave generates substantially larger velocities and hence the significant growth of the vortices induces the increase of the drag force. With the longer wavelength, the trapezoidal bar still experiences the largest drag force. However, the drag forces are comparable for the semicircular and triangular shapes. Figure 17b displays the numerical predictions of the lift forces. Analogous to the shorter wave, the considerably large lift force is observed for the case of the semicircular type because the velocities increase under the longer progressive waves. Additionally, the discrepancies

of the lift forces between the trapezoidal and triangular types increase. It is noted that the periodic oscillations of the lift force break to two small waveforms on the primary crests. This is related to the deformation of the waveform and hence generation of the higher harmonic waves.

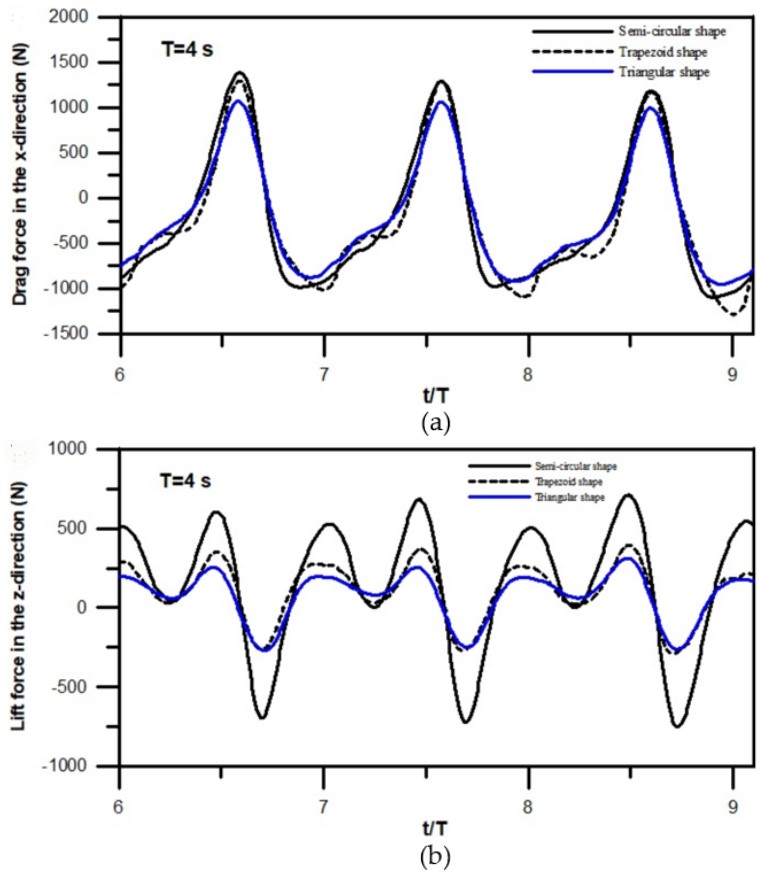

**Figure 17.** Simulation of the (**a**) drag and (**b**) lift forces for the different configurations of the submerged structures with the incident wave height $H_i = 0.2$ m and $T = 4$ s.

## 4. Conclusions

The air-water two-phase flow model was developed to study the linear surface waves interacting with the submerged structures. The combination of the LS method to track the surface motions and GCIB method with the developed weighted interpolation method to calculate the virtual forces inside the immersed boundary successfully solves the problem of the deformed surface and the corresponding flow fields around the immersed body induced by the wave transformation. The model demonstrates the accuracy via the validation with the experiment of the waves passing over the trapezoidal submerged breakwater [7].

The simulations of the waves over the trapezoidal bar show that the wave transformation is dependent on the wavelengths. Slight deformation of the wave profiles is observed for the short waves with the ratio of $L/qh_i = 14.6$ (T $= 1.01$ s) and hence the waves maintain the regularity. In contrast, for the longer waves with $L/qh_i = 57.7$ (T $= 3.03$ s), the shoaling effect significantly varies the waveprofiles to irregular and causes the large surface deformation associated with the wave energy distributed from the primary waves to the higher harmonics. Regarding the wave traveling over the submerged structures with the various configurations, the numerical results show that the considerable different patterns of the vortices are observed around the obstacles due to the structural types. The temporal and spatial features of the vortices are according to the location of the wave crests, resulting in the variations of the drag and lift forces, which significantly increase with the increase of the wavelength. For the simulated types of the semicircle, trapezoid, and triangle, the triangular submerged structure

experiences the smallest drag and lift forces. This is an interesting phenomenon which suggests a probable solution of the optimal design of submerged breakwaters. The present novel approach with the relatively simple algorithm and rigid grid system shows efficiency and high resolutions, which is in advantage of simulating the fluid-structure interaction for breakwaters acted on by the breaking waves as well as the support structure of wind turbines. Additionally, the promising results indicate the extension of the present 2D two-phase model to the 3D problems, for example, the dynamic responses of a moving body for the future studies.

**Author Contributions:** Y.-S.T. and D.-C.L. planned and carried out the present numerical experiment and writing, review, and editing the manuscript. D.-C.L. contributed the numerical computations and data analysis and writing and review the article. All authors have read and agreed to the published version of the manuscript.

**Funding:** This research was founded by the Ministry of Science and Technology, Taiwan, with the grant number No. MOST 109-2221-E-992-058 and MOST 108-2221-E-992-021.

**Acknowledgments:** The authors wish to express their grateful acknowledgement to the Ministry of Science and Technology, Taiwan, for founding this research.

**Conflicts of Interest:** The authors declare no conflict of interest.

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
