# Peer review of "A Ghost-Cell Immersed Boundary Method for Wave–Structure Interaction Using a Two-Phase Flow Model"

_water, doi:10.3390/w12123346_

Round 1
Reviewer 1 Report
This article contains interesting results from computer simulations and could be a valuable source for researchers in this area. The article is well-structured, contains the necessary substantive data and is written logically and consistently. In the opinion of the reviewer, it can be published, but the following comments require improvement and supplementation.
1. If we quote the N-S equation, it should be done with due diligence. In equation (2) (nabla u u) is misspelled, g is acceleration, f is force, and (Ab u) is velocity because Ab is dimensionless. A list of all physical quantities with their dimensions should be attached and the equations should be corrected so that the dimensions match.
2. The article presents the results of numerical research. The article should be supplemented with considerations on the possibility of experimental verification of the model and simulation.
Author Response
As attached file.

Reviewer 2 Report
I can not identify where is the novelty of this work. There are many works validating surface elevation and velocity fields for waves passing over submerged structures. What can Ghost-Cell Inmeresed Boundary Method add in comparison to other meshbased CFD methods based on finite volume... like OpenFOAM (one of the most open-source codes used in the field) or REEF3D, FLOW3D, etc....
What are the advantages of the method over meshless CFD methods like SPH where there is no need of an extra algorithm to detect the free surface?
What the method is two-phase? is the air phase key in the results here? I do not think so since there are no breaking waves or trapped air.
Authors wrote: "Besides, in use of LS and VOF method together, the algorithm is capable of predicting the complex phenomena such as air-water entrainments or wave breaking [27, 28]." However they do not study these problems in this work, right?
Can you please add measurements of error for your validation resutls and convergence study?
Results of this numerical approach should be also compared in terms of accuracy and efficiency with other CFD codes.
Figure 9 is missing in my PDF copy
I think that only one simple validation is not enough and not challenging at all. The work should include more validation cases.
Results need a better organisation, please.
What about computational resources?

Author Response
As attached file.

Reviewer 3 Report
This paper discuss a new numerical technique for computing two-phaseflows in which the level set method is employed to describe the motion
of the interface while a ghost-cell immersed boundary is used to
impose boundary conditions on immersed objects boundaries.
The manuscript is adequately presented and deserve publication after
the authors addressed the few technical comments in the following.
a) Figure.9 is missing.
b) Authors contributions to the numerical should be elaborated with more specifics in the introduction as well as into conclusions. Specifically, what is the state-of-the-art for such schemes and what the advantages over other methods? Do the computational results in this paper generate new knowledge regarding moderate Reynolds shallows water phenomenon? May the authors discuss more these aspects into introductions and conclusions?
c) Bibliography should be enhanced. There are several recent work on immersed boundary that should be acknowledged (among others https://doi.org/10.1016/j.compfluid.2018.12.014 , https://doi.org/10.1016/j.compfluid.2016.06.014)
Author Response
As attached file.

Reviewer 4 Report
The paper is already in an excellent form for publication.
Figure 9 is missing.
Author Response
As attached file.

Round 2
Reviewer 2 Report
The authors have been working hard to improve the paper and to address my previous comments. However several issues still need to be more developed such as:
1) Errors: I can not see the error estimations of the validation shown in figure 6 and 15, where the numerical results are compared with experimental data and with results obtained using Flow-3D. Please add those errors in terms of phase and amplitude. In that way you can support your analysis of the results better than only with visual comparison.
2) Computational runtimes of using Flow-3D should be also included in order to observe the higher efficiency running the proposed model here.
3) I do believe that authors should better highlight the advantages of the proposed numerical model including more information rather than a couple of 2 new sentences only in the introduction. In fact the advantages should be supported by the results in this paper and discussed also in the Conclusions section.
4) Conclusions should also include more examples of applications that can be now efficiently simulated with the proposed methodology rather than using more consuming CFD codes.
Author Response
As attached file.

Round 3
Reviewer 2 Report
Authors have finally modified the manuscript including more information related with all my comments. In my opinion the paper can be now accepted for publication in this journal.